# Reward Deficiency Syndrome (RDS) Surprisingly Is Evolutionary and Found Everywhere: Is It “Blowin’ in the Wind”?

**DOI:** 10.3390/jpm12020321

**Published:** 2022-02-21

**Authors:** Kenneth Blum, Thomas McLaughlin, Abdalla Bowirrat, Edward J. Modestino, David Baron, Luis Llanos Gomez, Mauro Ceccanti, Eric R. Braverman, Panayotis K. Thanos, Jean Lud Cadet, Igor Elman, Rajendra D. Badgaiyan, Rehan Jalali, Richard Green, Thomas A. Simpatico, Ashim Gupta, Mark S. Gold

**Affiliations:** 1Division of Addiction Research & Education, Center for Psychiatry, Medicine, & Primary Care (Office of the Provost), Graduate College, Western University of Health Sciences, Pomona, CA 91766, USA; dbaron@westernu.edu; 2Institute of Psychology, ELTE Eötvös Loránd University, 1075 Budapest, Hungary; 3Division of Nutrigenomics, The Kenneth Blum Behavioral Neurogenetic Institute, (Ivitalize, Inc.), Austin, TX 78701, USA; luisllnos522@gmail.com (L.L.G.); pathmedical@gmail.com (E.R.B.); rjalali@ivitalize.com (R.J.); rickgreen@newresourcesmedicalarts.com (R.G.); 4Department of Psychiatry, University of Vermont, Burlington, VT 05405, USA; thomas.simpatico@uvm.edu; 5Department of Psychiatry, Wright University Boonshoff School of Medicine, Dayton, OH 45324, USA; 6Column Health Clinic, Lawrence, MA 01843, USA; tmclaugh50@gmail.com; 7Department of Molecular Biology, Adelson School of Medicine, Ariel University, Ariel 40700, Israel; bowirrat@gmail.com; 8Department of Psychology, Curry College, Milton, MA 02186, USA; edward.modestino@gmail.com; 9Alcohol Addiction Program, Latium Region Referral Center, Sapienza University of Rome, 00185 Roma, Italy; mauro.ceccanti@uniroma1.it; 10Behavioral Neuropharmacology and Neuroimaging Laboratory on Addictions, Clinical Research Institute on Addictions, Department of Pharmacology and Toxicology, Jacobs School of Medicine and Biosciences, State University of New York at Buffalo, Buffalo, NY 14203, USA; pkthanos@gmail.com; 11Department of Psychology, State University of New York at Buffalo, Buffalo, NY 14203, USA; 12Molecular Neuropsychiatry Research Branch, DHHS/NIH/NIDA Intramural Research Program, National Institutes of Health, Baltimore, MD 21224, USA; jcadet@intra.nida.nih.gov; 13Center for Pain and the Brain (PAIN Group), Department of Anesthesiology, Critical Care & Pain Medicine, Boston Children’s Hospital, Boston, MA 02115, USA; dr.igorelman@gmail.com; 14Cambridge Health Alliance, Harvard Medical School, Cambridge, MA 02139, USA; 15Department of Psychiatry, South Texas Veteran Health Care System, Audie L. Murphy Memorial VA Hospital, Long School of Medicine, University of Texas Health Science Center, San Antonio, TX 78229, USA; badgaiyan@gmail.com; 16Department of Psychiatry, MT. Sinai School of Medicine, New York, NY 10003, USA; 17Future Biologics, Lawrenceville, GA 30043, USA; ashim6786@gmail.com; 18Department of Psychiatry, Washington University School of Medicine, St. Louis, MO 63110, USA; drmarkgold@gmail.com

**Keywords:** dopamine, hypodopaminergia, Genetic Addiction Risk Severity (GARS) Test, pro-dopamine regulation (KB220)

## Abstract

Reward Deficiency Syndrome (RDS) encompasses many mental health disorders, including a wide range of addictions and compulsive and impulsive behaviors. Described as an octopus of behavioral dysfunction, RDS refers to abnormal behavior caused by a breakdown of the cascade of reward in neurotransmission due to genetic and epigenetic influences. The resultant reward neurotransmission deficiencies interfere with the pleasure derived from satisfying powerful human physiological drives. Epigenetic repair may be possible with precision gene-guided therapy using formulations of KB220, a nutraceutical that has demonstrated pro-dopamine regulatory function in animal and human neuroimaging and clinical trials. Recently, large GWAS studies have revealed a significant dopaminergic gene risk polymorphic allele overlap between depressed and schizophrenic cohorts. A large volume of literature has also identified ADHD, PTSD, and spectrum disorders as having the known neurogenetic and psychological underpinnings of RDS. The hypothesis is that the true phenotype is RDS, and behavioral disorders are endophenotypes. Is it logical to wonder if RDS exists everywhere? Although complex, “the answer is blowin’ in the wind,” and rather than intangible, RDS may be foundational in species evolution and survival, with an array of many neurotransmitters and polymorphic loci influencing behavioral functionality.

## 1. Introduction

### 1.1. Summary Preamble

Bob Dylan’s protest song, “Blowin’ in the Wind,” written in 1962, poses rhetorical questions about peace, war, and freedom. The song suggests that the intangible, impenetrable, and ambiguous answers are either “blowin’ in the wind” or evident. This review suggests that the “answer” Reward Deficiency Syndrome (RDS) is apparent everywhere, that RDS might have been essential to human evolution and survival, and that it may lead to a molecular neurogenetic understanding of pain sensitivity and many mental health disorders that affect mood, thinking, and behavior.

### 1.2. “Blowin’ in the Wind”—Bob Dylan, 1962

Reward Deficiency Syndrome (RDS) refers to many mental health disorders. These illnesses include a wide range of addictions and compulsive and impulsive behaviors. RDS is like an octopus of behavioral dysfunction. While it is not as yet in the DSM, RDS refers to the breakdown of reward neurotransmission and the destructive behaviors initiated by the combination of environmental (epigenetic) influences and DNA-based neurotransmission deficits that interfere with the usual achievement of the satisfaction of human physiological drives (food, water, sex). The basis of the RDS concept was the initial finding of Blum et al. [1] that the A1 (minor) allele of the D 2 dopamine receptor (DRD2) gene was associated with severe alcoholism, and later associations with smoking, obesity, and other substances and non-substance-disordered behaviors were identified [2,3]. 

This article aims to explore the RDS behavioral model and the potential for treatment it may provide. As a featured psychological disorder, RDS [4] is now the subject of global research, with 213 listings in PubMed (1-4-22) and, using the word term “reward deficiency”, 1409 listings. Is the answer “blowin’ in the wind”?

## 2. Genes and Evolution for the Survival of Our Species

We know that some neurotransmitter gene variants have been present since the beginning of time. Charles Darwin and others have examined the basic principles of Biological Science. They investigated the mechanisms that steer behavior and biological development. The theory on natural selection, particularly the sexual selection process, gained significance during the 19th century, especially when asking, “what is being human?” The capacity for natural selection and evolution are not human accomplishments or a sign of our uniqueness. We humans, it seems, are ingenious in fooling ourselves and others, especially when we are in love or desperately in search of it. The conjecture of modern biological theory established that organisms result from evolutionary competition, and genes that propagate the fittest phenotype and behaviors and generate offspring will best survive [5]. 

Reward behaviors that generate learning, motivation, economic decisions, and positive emotions support survival, mating, and care for offspring, ensuring the reproduction and survival of the organism. Homo sapiens can live well enough to reproduce, survive, and propagate their genes because foods and drinks are powerful rewards. Information and economic exchanges ensure sufficient palatable food and drink supply, while powerful sexual attraction supports mating and gene propagation. David E. Comings suggested that better care for offspring might result in an evolutionary reproductive advantage [6]; additional reward mechanisms, such as novelty seeking and exploration, widen the spectrum of available rewards and, thus, enhance the chance for survival, reproduction, and ultimately gene propagation [7]. 

Theories about pleasure being a salutogenic factor for health promote positive feelings, satisfaction, and happiness, and result from the anticipation of an enjoyable experience, possibly through dopamine release [8]. The reward function in gene propagation and evolutionary fitness defines the immediate everyday behavioral reward functions, such as the pleasures involved in eating, drinking, mating, and nurturing offspring [7].

Pleasurable activities may stimulate personal growth and healthy behavioral changes, including stress management. Esch and Stefano proposed that the brain reward system connects compassion and love. Pleasure induction suggests that social contact, love, attachment, and compassion can be highly effective in stress reduction, survival, and overall health [9]. 

These natural rewards, essential for survival and appetitive motivation, lead to beneficial biological behaviors, while experimentation with addictive substances and behaviors can act directly on reward pathways, causing the deterioration of these systems and promoting hypodopaminergia. The role of neurotransmission and both positive and negative pleasurable states have been studied over many decades [10,11,12,13,14,15,16,17,18,19]. Although new studies that investigate the anatomical and neurobiological functions of animals and Homo sapiens are required, some of the known complex neurobiological limbic activity of the “Brain Reward Cascade” (BRC), involving endorphin and endogenous opioidergic and other neurochemical mechanisms, are illustrated in Figure 1 [20,21].

The impact of evolution on the genetic makeup of Homo sapiens in terms of survival must also be considered. Genes related to the dopaminergic system, as an example, and the impact of epigenetics on behavior, including addiction liability [22], may have served as an important survival tactic, but in our current society may no longer serve survival. Carriers of DNA polymorphisms, such as the DRD2 Taq A1 allele, with behaviors such as competitiveness and selfishness, may have had survival advantages as hunter–gatherers millions of years ago, but in today’s world may be counter to achieving positive relationships and happiness [23]. This conundrum may advance the unfortunate condition that Homo sapiens are wired to seek pleasure in all its forms, no matter the cost to oneself, others, and one’s surroundings.

## 3. Reward Deficiency Syndrome

Empirical evidence from a large amount of data demonstrates overlaps in the etiology, phenomenology, and the underlying psychological and biological mechanisms of different types of addictive behavior. There has been a shift in the conceptualization of addictions to include non-substance (previously process) addictions. Kotyuk et al. [24] performed an epidemiological analysis of data collected from adolescents and young adults, N = 3003 (42.6% males; mean age 21 years), as part of the Psychological and Genetic Factors of the Addictive Behaviors (PGA) study. Addictions to psychoactive substances and behaviors were rigorously assessed. The study provided lifetime data on substance use, co-occurrences, and prevalence estimates of specific behavioral addictions. Koyuk et al. identified associations between (i) smoking, gambling, problematic Internet use, exercising, and eating disorders, (ii) alcohol consumption, problematic Internet use, gaming, and eating disorders, and (iii) cannabis use, problematic online gaming, and gambling [24]. As suggested by the authors, these results indicated a considerable overlap between the occurrence of these addictions and behaviors and accentuated the likelihood that these disorders have common psychological, genetic, and neural pathways. The RDS concept and the addiction component model propose a common phenomenological and etiological background for addictive, impulsive, and compulsive behaviors.

## 4. Prevalence of Addiction-Related Gene Polymorphisms

This convincing evidence of the co-occurrence between drug and non-drug addictive behaviors supports the RDS concept. There are an array of addiction-related gene polymorphisms, currently up to over 39,632, in which both the glutamate and dopaminergic pathways are very prominent. Various ethnic groups carry differential prevalence for each particular allele; Table 1 shows some examples of widespread general population prevalence.

The variants included in the Genetic Addiction Risk Severity (GARS) test are the most common, and are studied more in people of European descent. Some variants are more common in certain ethnicities and less prevalent in others. Table 1 shows the global heterozygous prevalence of risk variants in the general population. They include: DRD2 46%, D RD3 41%, DRD4 42%, OPRM1 29%, and 5HTTLPR 43%. Table 1 also displays the percentage of subjects with variants, as measured in one testing center, derived from a mixed-gender population of seven chemical dependence treatment centers across the United States. The population diversity percentages for homozygous (single nucleotide polymorphisms (SNP), homozygous normal, and heterozygous SNPedia provided the global heterozygous prevalence of risk variants in the following populations (rs4532; rs1800497; rs6280; rs1800955; rs4680, and rs1799971) [25].

To elaborate on the notion that RDS is, surprisingly, found everywhere in Homo sapiens, we have predicated this conclusion based on the high percentage of prevalence found throughout the small amount of literature described above [26]. Individuals with mood disorders or addiction, impulsivity, and some personality disorders can share a dysfunction in how the brain perceives reward, where the processing of natural endorphins or the response to exogenous dopamine stimulants is impaired [27,28]. Reward Deficiency Syndrome is a polygenic trait with implications that suggest cross-talk between different neurological systems, including the known reward pathway, neuroendocrine systems, and motivational systems. Results from studies using animal models of substance use disorder (SUD), major depressive disorder (MDD), early life stress, immune dysregulation, attention deficit hyperactivity disorder (ADHD), post-traumatic stress disorder (PTSD), compulsive gambling, and compulsive eating disorders, as well as anhedonia, found underlying reward deficiency mechanisms in multiple brain centers. The widespread and remarkable array of associated/overlapping behavioral manifestations with the common root of hypodopaminergia, the basic endophenotype of these disorders, comprehended as RDS, has indeed been likened to a behavioral octopus [26]. In this regard, previous work from Blum’s group provides the framework within which to suggest that the true phenotype is RDS, and behavioral disorders are endophenotypes.

Abnormal behaviors involving dopaminergic gene polymorphisms often reflect an insufficiency of usual feelings of satisfaction. A family history of SUD may indicate an inherited neurotransmitter dysfunction. Exposure to periods of chronic stress or alcohol or other substances can, via epigenetic mechanisms, compound allele-based reward neurotransmitter dysfunction [29,30]. An experimental group of n = 55 subjects derived from up to five generations from two families and a group of rigorously screened control subjects (n = 30) were tested for the A1 allele of the DRD2 and the DAT1 gene and the associated risk allele. These subjects, including 13 deceased family members, provided data related to RDS behaviors. The RDS risk alleles were found significantly (*p* < 0.015) more often in the RDS families vs. controls. The TaqA1 allele occurred in 100% of Family A individuals (N = 32) and 47.8% of Family B subjects (11 of 23). We concluded that although our sample size was limited and linkage analysis was necessary, the results supported the role of dopaminergic polymorphisms in RDS behaviors. This study demonstrated a non-specific RDS phenotype and showed how evaluating single subset behaviors of RDS might lead to spurious results. The acceptance of a non-specific “reward” phenotype may help future GWAS and association and linkage studies involving reward polymorphisms and other neurotransmitter gene candidates [31].

## 5. The Reward Deficiency Syndrome (RDS) Model

The RDS model emphasizes a spectrum of biological systems approach for assessing relationships among different behaviors, integrating psychological, neurological, and genetic factors of addictive, impulsive, and compulsive behaviors. The RDS model, introduced in 1995 [29,30], proposes that RDS may have an important role in addictive behaviors, such as alcoholism, polysubstance abuse, smoking, and gambling, impulsive disorders, such as attention-deficit hyperactivity disorder, compulsive behaviors, such as compulsive sexual behaviors and overeating, and personality disorders, such as conduct or antisocial personality disorders. The RDS model proposes that a deficiency in reward system functioning may emerge due to molecular, genetic, and environmental effects. 

A hypothesized hypodopaminergic state relates to motivations to engage in risk-taking behaviors, substance use, or other potentially addictive behaviors to address a dopamine-related deficiency in a reward system. Thus, individuals with RDS would be characterized by low levels of dopaminergic functioning, motivating them to engage in addictive behaviors. The model proposes that lower dopamine levels trigger individuals to be less satisfied with natural rewards (a blunted response at the brain reward circuitry) and motivated to engage in substance use or other risky or addictive behaviors. Individuals may experience short-term gratification and eventually or quickly develop chronic substance use or addictive disorders. Similarly, they may overeat or practice extreme sports, such as car racing, to stimulate their dopamine systems externally. Other molecular and environmental factors that are not yet fully understood may delineate the specific type of behavioral disorder engaged in by these individuals. In summary, the RDS model proposes psycho-genetic markers of several impulsive–compulsive or addictive behaviors that focus on rewards and dopamine systems.

## 6. The Neuro-Genetic Background of the RDS

### 6.1. Neurotransmitters and Mental Illness

In 1956, the doctrine of Jellinek shocked the world when he proposed the concept of alcoholism as a disease without much scientific evidence [32]. This idea was not generally accepted. However, scientists at that time agreed, in part, that deficiencies or imbalances in brain chemistry-perhaps genetic in origin, contributed to the cause of alcoholism and other addictions [33]. In the early 1960s, the interrelatedness of reward circuitry and the prefrontal cortices of the brain was not well understood. The core neurotransmitters were unknown. Serotonin, GABA, dopamine, acetylcholine, were not well characterized, and endorphins were not a part of the scientific acumen.

### 6.2. Genetic Associations

Blum’s group identified and published the first gene associated with severe alcoholism [1]. To date, there have been many studies using case–controls. Nonetheless, adequate RDS-free controls have not been used. This problem needs to be addressed and, if clean controls are appropriately adopted in the addiction field, spurious results would be eliminated, and confusion regarding the genetics of addiction reduced. 

Can SUD be better treated using early genetic risk screening, enabling early intervention by the induction of dopamine homeostasis? The conceptualization of RDS identifies the neurogenetic basis of all drug and non-drug addictive, obsessive, impulsive, and compulsive behaviors characterized by causal pathways and mechanisms of genetic dysfunction that manifest as behavioral and psychological characteristics after initiation by epigenetic conditions.

After the initial conceptualization of RDS by Blum in 1995, Blum et al., using the Bayesian Theorem [34], found that carriers of the DRD2 A1 allele had a predictive value (PV) for any future RDS behaviors, such as substance use disorder (SUD), obesity, and shopping addiction, that reached 74.4% [34]. The number of genes known to be involved in reward has increased significantly over time. Blum et al. have selected eleven alleles from ten reward genes from the extensive literature for the Genetic Addiction Risk Severity (GARS) test [35,36,37,38].

However, based on results from a sampling of case–control studies, an estimation found significant associations for alcohol and drug risk. A total of 110,241 cases and 122,525 controls derived from the current literature strongly suggest that despite concerning controls (e.g., blood donors), it is remarkable that there are so many case–control studies that indicate selective associations with the risk alleles which indicate hypodopaminergia, measured by the GARS test.

## 7. Phenomenological Aspects of the RDS

Although the molecular mechanisms of RDS are phenomenological, the classification of its appearance is relatively incomplete. There are no available unified definitions of the psychological and behavioral appearance of RDS. The proposed RDS includes a set of psychological “symptoms” that can signal its presence. Blum and colleagues refer to the phenomenological and behavioral aspects of the RDS as “an inability to derive reward from ordinary, everyday activities” [36,37]. Dopamine, as well as additional reward neurotransmitters, are portrayed as producing this sense of well-being. People with neurotransmitter dysfunctions engage in substance seeking and craving behavior and employ other common hedonic mechanisms to reduce negative emotion [31,39,40]. 

The RDS model proposes that insufficiencies in dopaminergic systems render individuals susceptible to addictive behaviors through the stimulation of the mesolimbic system. Based on the phenomenological appearance of RDS, the purportedly linked disorders and behaviors, and the proposed involvement of the mesolimbic system, RDS would theoretically show relatedness to risk-taking personality traits, such as impulsivity and novelty seeking, as well as mood characteristics, such as anhedonia or depression. In summary, the multi-level model of RDS describes a so-called “hypodopaminergic trait”, which associates with psychological dimensions of addictions and potentially addictive behaviors and proposes a specific molecular mechanism. The exception may be most adolescents because of developmental epigenetics, which may induce a hyperdopaminergic state [41].

## 8. Evaluation of the RDS Model

Despite the promise of the RDS model, some of the proposed associations have received mixed support, and the model needs further empirical testing. For example, as noted above, one of the initial premises of the RDS based on early findings—the relationship between DRD2 variants and addictions—has been questioned [42], or has shown small effect sizes [43]. Others found that the A1 allele does not increase the risk for alcoholism per se, but may be involved in related traits or characteristics [44]. Inconsistent association results involving DRD2 variants and addictions suggest more complex etiologies for addictions. These questions were first raised over 20 years ago [45,46,47,48,49,50], but currently, there is general agreement that addictions are polygenetic, and that the DRD2 variants represent a major polymorphic allelic concern [50].

Empirical studies also question the link between RDS and food addiction. Benton and Young [51] conducted a meta-analysis of BMI and DRD2 variants to test the hypothesis stating that, similar to substance use disorder (SUD) in food addiction, the A1 allele is associated with lower levels of DRD2 genes [52,53]. This meta-analysis of 33 studies found no associations for body mass index (BMI), which they criticized as a definitive measure of food addiction [54]. They concluded wrongly that this meta-analysis did not support the RDS model of obesity or food addiction. However, many of the studies assessed in this meta-analysis suffered from inappropriate food addiction severity phenotypes and a lack of stratification among racial groups. These referenced studies conversely show the involvement of dopamine genetics in food seeking [55,56,57,58,59,60,61,62,63,64,65,66,67,68,69,70,71,72,73,74,75,76,77]. 

Leyton [78] argued that increased dopamine levels equating to pleasure (as often appears in the RDS literature) is an outdated view. More recent findings show that increased dopamine may motivate approach, meaning that stimuli can draw and sustain motivational salience. He also argues that increased rather than decreased dopamine is precipitated by drug-seeking behavior. Some neuroimaging studies explored the opponent process (lower DRD2 receptors), whereby A1 carriers showed a smaller blunted interference effect to reward alone, with A2 homozygotes exhibiting a specific interference reduction during combined reward and punishment trials [79,80]. Many neuroimaging studies display a blunted response to cue activity at the brain reward circuitry in the presence of the DRD2 A1 allele. Stice et al. [81] found that cross-sectional and prospective data from two functional magnetic resonance imaging studies imply that individuals may overeat to compensate for a hypo-functioning (blunted) dorsal striatum, particularly those with genetic polymorphisms (such as the DRD2 A1 allele) thought to attenuate dopamine signaling in this region.

Others have questioned the centrality of dopamine to addictions. Nutt et al. reviewed the origins of the dopamine theory of addiction and the ability of addictive drugs to elicit dopamine release in the human striatum [82]. They found evidence that alcohol (moderately) and stimulants (robustly) increase striatal dopamine levels, but found little evidence that cannabis and opiates increase dopamine levels [82]. However, van der Kooy’s group proposed that opiate reward processes and the mesolimbic dopamine system have functional control of the motivational state. They determined that D2 receptor function is only crucial in mediating the motivational effects of opiates when the opiate-dependent animal is in a withdrawn motivational state. These findings also underscore the important influence of the genetic background of a given phenotype, evidenced when morphine place preference is abolished in C57Bl/6 wild-type mice by D2 receptor knockout [83,84,85]. 

Stringer et al. [86] carried out a meta-analysis of data from a sample of 32,330 subjects from the International Cannabis Consortium regarding dopamine. They found a strong genetic correlation between lifetime cannabis use and cigarette smoking (rg = 0.83; *p* = 1.85 × 10^−8^) that implies that the SNP effect sizes of the two traits are highly correlated. They recommended four genes (NCAM1, CADM2, SCOC, and KCNT2) as candidates for follow-up functional studies, and suggested an examination of the functional role of the NCAM1 gene in combination with other genes in the same gene cluster (NCAM1–TTC12–ANKK1–DRD2) [86].

Although Leyton [78] argues that increased rather than decreased dopamine precipitates drug-seeking behavior, several studies counter his view [50]. Based on decades of data, especially in gambling disorders [87,88], others have questioned dopamine’s centrality to addictions. However, recent global data suggest otherwise [89,90,91,92,93,94,95,96,97,98].

Volkow et al. [99] used positron emission tomography (PET) to corroborate drug-induced fast dopamine (DA) increases in the human striatum, including nucleus accumbens (NAc). However, they unexpectedly found diminished drug-induced DA increases (as well as their subjective reinforcing effects) in addicted subjects compared to controls. In contrast, addicted subjects showed significant striatum DA increases in response to drug-conditioned cues associated with self-reports of drug cravings that were greater than the DA responses to the actual drug. Volkow et al. [99] postulated that the discrepancy between the expectation for the drug effects and the blunted pharmacological effects helps maintain chronic substance use due to attempting to achieve the expected reward (conditioned response). Additionally, addicted subjects tested early or during protracted withdrawal have lower levels of D2 receptors in the striatum (including NAc). During withdrawal, these lower levels of D2 receptors are associated with decreases in baseline activity in the frontal brain regions implicated in salience attribution (orbitofrontal cortex) and inhibitory control (anterior cingulate gyrus), whose disruption results in compulsivity and impulsivity and potential drug reinstatement or relapse. This observed imbalance between dopaminergic circuits that underlie reward and conditioning and those that underlie executive function (emotional control and decision making) contribute to compulsive drug use and loss of control in addiction [99]. It is not surprising that drug-dependent subjects show a blunted brain reward circuitry response to psychoactive drugs or stimuli due to genetic antecedents, such as, for example, the DRD2 A1 allele, and show impaired dopamine function. Work by Ariza et al. [100] suggested that the A1 allele of the DRD2/ANKK1-TaqIA gene is associated with addictive disorders, obesity, and the performance of executive functions. To help us understand Volkow et al.’s [99] so-called surprising results related to blunted executive function, it is, indeed, quite remarkable that Ariza et al. [100] found a genetic deficit related to ‘DRD2/ANKK1-TaqIA (previously shown to reduce the number of D2 receptors [52,53]). Specifically, the ‘DRD2/ANKK1-TaqIA A1 allele status significantly affected almost all the executive variables conferring a blunted performance of executive functions [99].

While the RDS model may represent an important framework for considering multiple addictions, multiple additional psychological and molecular factors may constitute important mechanisms. Some have argued that although reductionist attempts to try to explain complex behaviors with simple elements are essential, they may not be sufficient, emphasizing the complex nature of addictive behaviors, reward mechanisms, and related neural mechanisms [99,101,102]. MacKillop and Munafò [103] propose that empirical data and theoretical models highlight the need for an “intermediate phenotype” approach to addictions. This approach states that endophenotypes putatively “connect the dots” from the genome to the syndrome [103], which share some features with what Blum and colleagues proposed in the RDS model. Specifically, the Genetic Addiction Risk Severity (GARS) test may define a genetic pattern associated with RDS, constituting a risk phenotype for addictions in general. However, further empirical studies of the RDS model are needed to answer whether RDS is an adequate phenotype for addictions or not.

Methodologically, some reviews summarizing studies supporting the RDS model are selective rather than systematic reviews [104]; however, others have attempted to provide an independent review of RDS [105]. Moreover, the controversy surrounding the association of the DRD2 polymorphisms and generalized substance use disorder (SUD) was informed by the independent reviews from Lopez-Leon et al. [106], Deak and Johnson [107], and Plamer et al. [108].

It is noteworthy that other work from Goldman’s laboratory further supported the role of the DRD2 Taq A1 allele in alcohol use disorder (AUD). Jung et al. [109] executed a meta-analysis involving a total of 62 studies of DRD2 and AUD. Participants (n = 16,294) were meta-analyzed. The rs1800497 SNP was associated with AUD (with an odds ratio, 1.23; 95% CI, 1.14–1.31; *p* < 0.001). Jung et al. [109] stated that the association was attributable to spuriously low allele frequencies in controls in positive studies, which also accounted for some between-study heterogeneity. While this sounds negative, it is indeed the opposite. The use of the term “spuriously low allele frequencies in controls” is, instead, very positive. A low rs1800497 SNP in controls attributed to 43% of the variance is a recommendation from an earlier article from Blum et al. that carefully discusses the importance of having low allele frequencies of the rs1800497 SNP, and which argues that having other risk alleles present in controls is akin to comparing disease with diseased, not clean controls. The holy grail in genetic testing is to remove any genetic risk polymorphism from controls since, unlike one gene–one disease (OGOD), RDS is polygenetic and very complex. In addition, eliminating any RDS-related behaviors from the control group would help obtain the best possible statistical analysis, instead of comparing the phenotype with unscreened controls [110].

Godman’s group [109] also said that a human postmortem brain analysis of expression quantitative loci in public data revealed that cis-acting loci perturb the DRD2 transcript level; however, rs1800497 does not and is not in strong disequilibrium with such a locus. No evidence has emerged that rs1800497, located in ANKK1, perturbs the expression or function of DRD2. However, previous research from Blum et al., laboratory, and others reported significant reductions of up to 30–40 percent of the DRD2 receptor in human subjects carrying the DRD2 A1 allele (either heterozygote or homozygote) compared to non-carriers of the rs rs1800497 allele [53]. Specifically, the binding characteristics (Kd (binding affinity) and Bmax (the number of binding sites)) of the D2 receptor antagonist as the ligand have been studied [53]. The adjusted binding affinity was significantly lower in alcoholic than in nonalcoholic subjects. For subjects with the A1 allele, there was a high association with alcoholism and a significantly reduced Bmax compared with subjects with the A2 allele. Subjects independent of alcoholism demonstrated a progressively reduced Bmax. The A2/A2 had the highest mean values, and the A1/A1 alleles the lowest. The differential expression of polymorphic dopamine receptor genes suggests dopaminergic system involvement in conferring susceptibility to severe alcohol use disorder. Thompson et al. [52] investigated adult middle-aged to elderly subjects with no psychiatric or neurological disorders concerning DRD2 expression, and provided additional support for this finding. Utilizing autoradiography in the caudate, putamen, and nucleus accumbens, using the specific D2 receptor ligand [3H]-raclopride, they found DRD2 expression in a sample of 44 individuals. The presence of one or two A1 alleles was associated with altered D2 receptor binding in all areas of the striatum, reaching statistical significance in the ventral caudate and putamen (*p* = 0.01 and *p* = 0.044, respectively). In 2020, work by Stanfill and Cao [111] found several haplotype DRD2 variants showing reduced DRD2 expression, including the rs19891549 and rs6277 known to be in linkage disequilibrium with rs10891549, and also with other well-studied variants Taq1A (rs1800497) located in the promoter region of the ANKK1 gene and -141C ins/del (rs1799732).

A recent systematic review and meta-analysis has robustly identified associations between genetic variation in dopaminergic receptor D2 and AUD (rs4936277 and rs61902812) and problematic alcohol use (rs138084129 and rs6589386) and gene-based associations with AUD, as indexed by Alcohol Use Disorders Identification Test (AUDIT) scores. The rs4936277 was associated with SUD at *p* ≤ 5 × 10^−9^, supplying robust association data [112].

Although a spectrum biological systems approach has value in conceptualizing and investigating addictions, the processes proposed in the RDS model have considerable potential; further empirical studies to evaluate the RDS theory and clarify proposed aspects, particularly in the setting of mixed or negative results, seems prudent.

## 9. The RDS Model: Clinical Applications

There are anecdotal reports from clinicians treating thousands of patients with mental health disorders suggesting a very high overlap between psychiatric disorders, major depression, and several personality disorders that, at first glance, seem to be independent psychological constructs, but which do fit the RDS model. In support of this proposal, there have been genetically based clinical studies revealing significant mental health problem overlaps related to dopaminergic gene polymorphism, including schizophrenia [31,45,48,112,113,114,115,116,117,118,119]. It is also valuable to consider the association of RDS with pain-induced anti-reward symptomatology and pleasure [11,120,121,122,123,124,125].

One of us (TM), when reviewing the medical charts of approximately 250 patients over seven years, found that 85% of opiate-addicted patients had a history of ADD or ADHD (an RDS sub-type), as did about 75% of those complaining of “depression,” and the vast majority responded affirmatively to two simple questions: (1) Did you have difficulty paying attention in grade school? (2) Do you spend time lying in bed for many hours in a row, like ten (or lying on the couch)? The latter may reflect amotivational syndrome, which is highly associated with ADHD. Other questions which may elicit this syndrome include: Do you feel like you are just going through the motions of life? Or that you just don’t care about anything? Given the high prevalence of amotivational syndrome in this population, one reason that anti-depressants are only 60% successful in treating major depression may be that by prescribing selective serotonin reuptake inhibitors (SSRIs) and serotonin–norepinephrine reuptake inhibitors (SNRIs), hypodopaminergia is not targeted. A test to uncover a genetic basis for the patient’s symptoms is indicated, especially in the drug-addicted population, with a history of ADD/ADHD and a personal and familial history of RDS behaviors. Additional questions which may well be helpful are “Do you whistle, hum, or talk to yourself, or do have a tendency to clear your throat or your nose (all vocal tics) or crack your bones or twist your neck, back, or hair?” (motor tics). Affirmative answers to the latter can point to Tic Disorder or Tourette’s Syndrome, both of which fall under the RDS umbrella.

The psychiatric history should include questions about substance use and physical, sexual, verbal, or emotional abuse histories. Familial evidence of RDS concerning reports of pathological aggression, incest, verbal abuse involving swearing (vocal tics such as the F-bomb and the C-word constitute vocal tics) are associated with Tic Disorder or Tourette’s Syndrome. The trauma induced by such abuse or neglect is expressed in PTSD symptomatology (an RDS subtype) and other psychological symptoms and diagnoses.

In addition, a detailed history of early attentional problems in grade school, frequently associated with procrastination and boredom, supports the diagnosis of ADD/ADHD and amotivational syndrome. Recently, increased attention has been devoted to undiagnosed and untreated ADD/ADHD in seniors by clinicians at the NYU Lagone Adult ADHD clinic.

In the authors’ experience, even casual conversation involving the RDS-informed clinician with non-patient friends and acquaintances can reveal the widespread prevalence of RDS subtypes. A future report will compare the prevalence figures of RDS subtypes (see Figure 2).

## 10. Mental Illness and RDS

Mental health disorders are considered to be health conditions that involve changes in emotion, thinking, or behavior. The origin of the condition distinguishes mental health disorders from mental illnesses. Mental illnesses are considered to be bodily diseases diagnosed by a medical professional that interfere significantly with cognitive, emotional, or social abilities. Mental illnesses, including mood disorders, such as depression, anxiety, and bipolar disorder, psychotic disorders, such as schizophrenia and eating disorders, and psychiatric conditions, such as personality disorders, occur with varying severity.

Presently the definitive resource for all mental disorders, *The Diagnostic and Statistical Manual of Mental Disorders* (DSM), includes diagnostic criteria for substance use disorders that distinguish between two types: substance use disorders and substance-induced disorders. Substance use disorder criteria hinge on the harmful consequences of repeated use, but substance-induced disorders include intoxication, compulsive use, tolerance, or withdrawal. Concomitantly, genetic covariance among substance- and non-substance-disordered individuals can be due to individuals carrying reward gene allele variations (polymorphisms). Thousands of studies, reviews, meta-analyses, and cases, albeit with the need for additional required studies, show significant dopaminergic gene polymorphism overlaps between many psychiatric illnesses and RDS (see Figure 2). This genetic covariance among individuals, due to individuals carrying gene allele variations, can be used to predict phenotypic responses to environmental circumstances.

## 11. Conclusions

The large volume of literature discussed here underpins the known neurogenetic and psychological parameters of RDS. It supports the hypothesis that substance use disorders, such as OUD, AUD, and overeating, non-substance use disorders, such as gambling, and other behaviors, such as PTSD and ADHD, are endophenotypes, with the true phenotype being RDS, [31] see Figure 3.

While it is indeed rhetorical, RDS is ubiquitous, and while both complex and evident, “the answer” is also apparent. Dylan wrote the song about major social issues, including war, peace, and racism. We propose that the dopaminergic deficiency central to RDS may indirectly link these cataclysmic societal events. Previously, we discussed this theory as it related to the molecular genetics of the holocaust [127]. However, the circumstance that RDS is everywhere does not mean that it is as intangible as the wind; instead, it links to our species’ evolution and survival.

## 12. Patents

KB is the inventor of both GARS and KB220 (Restoregen). The Restoregen precision patented formulation is exclusively licensed to Ivitalize Inc. ERB, EJM, LLG, RJ, and RG are unpaid members of The Kenneth Blum Behavioral and Neurogenetic Institute. There are no other conflicts to report.

## Figures and Tables

**Figure 1 jpm-12-00321-f001:**
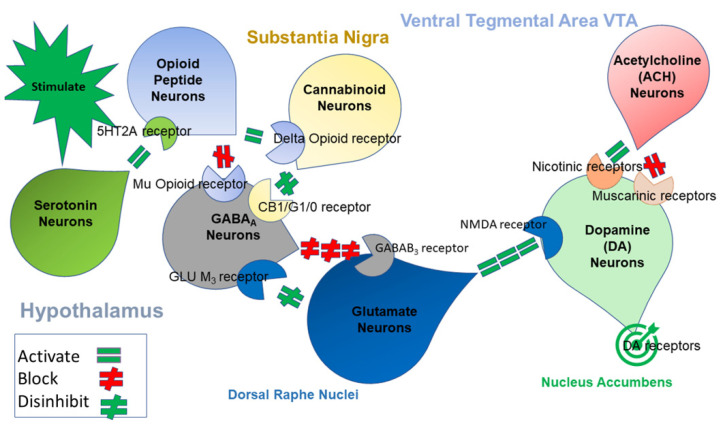
This cartoon figure illustrates some of the interactions of several neurotransmitter pathways in the Brain Reward Cascade (BRC) system. Environmental stimulation in the hypothalamus causes the release of serotonin, which, in turn, via, for example, 5HT-2a receptors activates (green equals sign) opioid peptides, releasing them into the hypothalamus. The opioid peptides have two distinct effects via two different opioid receptors. The first inhibits (red hash sign) through the mu-opioid receptor (possibly via enkephalin), which projects to the substantia nigra GABAA neurons, and the second stimulates (green equal sign) cannabinoid neurons (e.g., 2-archydonoglcerol and anandamide) through beta-endorphin-linked delta receptors, which, in substantia nigra, inhibit GABAA neurons. Activated cannabinoids, mostly 2-archydonoglcerol, can also disinhibit (red hash sign) substantia nigra GABAA neurons indirectly by activating G1/0-coupled CB1 receptors. Similarly, dorsal raphe nuclei (DRN) glutamate neurons can disinhibit substantia nigra GABAA neurons indirectly through the activation of GLU M3 receptors (red hash sign). Stimulated GABAA neurons will powerfully (red hash signs) inhibit VTA glutaminergic drive via GABAB 3 neurons. VTA glutamate neurons project to dopamine neurons through NMDA receptors (green equals sign), where they preferentially release dopamine at the nucleus accumbens, shown as a bullseye, indicating euphoria (a motivational response).

**Figure 2 jpm-12-00321-f002:**
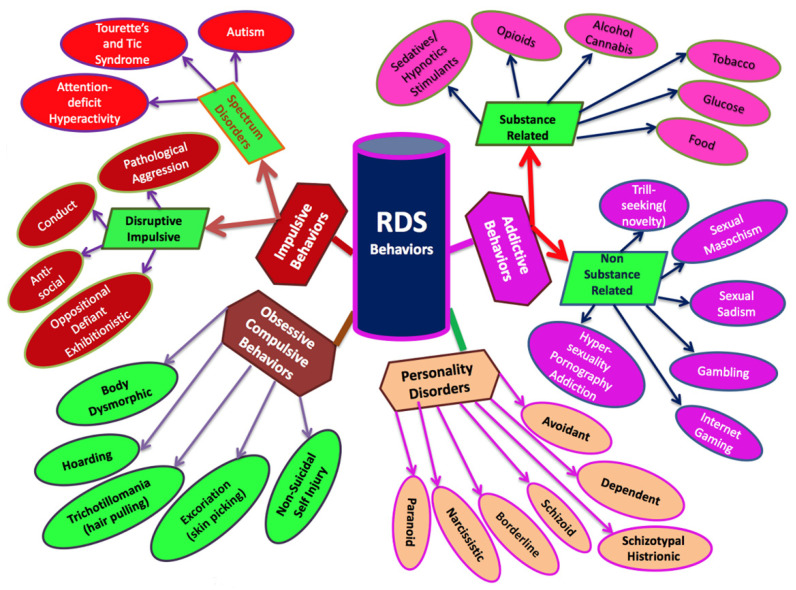
A schematic of RDS. This figure illustrates the categorization of RDS behaviors, showing which behaviors are addictive, impulsive, obsessive, and personality disorders. Sub-categories divide addictive behaviors into substance and non-substance related and impulsive into disruptive/impulsive and spectrum disorders. These behaviors have dopaminergic dysfunction in common: acute excess or chronic deficit of dopamine release in the brain reward circuitry.

**Figure 3 jpm-12-00321-f003:**
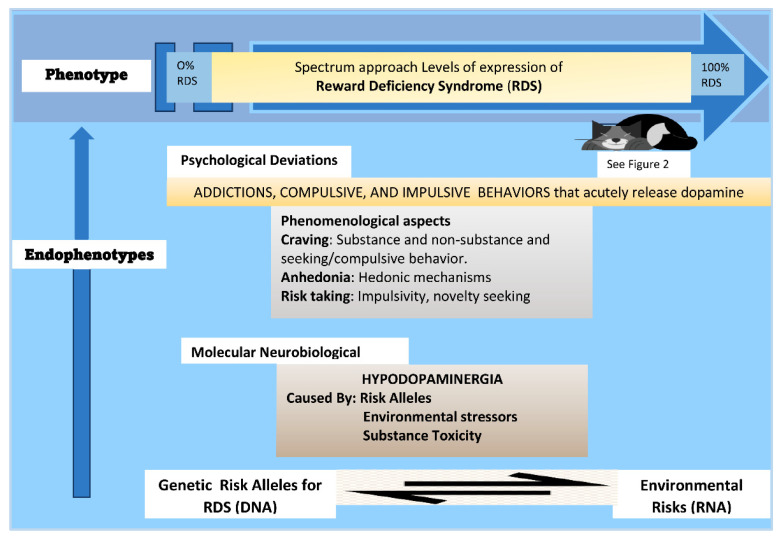
Summary of RDS Phenotype. This illustrates the relationship between endophenotypes and phenotype in RDS (modified from [126]).

**Table 1 jpm-12-00321-t001:** The Global Heterozygous Prevalence of Risk Variants in the General Population.

Gene with Most Common Risk Allele	Global Heterozygous Prevalence (%)	Percent of Subjects with Variant *
Dopamine D2 Receptor (DRD2): ** rs1800497—risk allele A1	46	Asian 33%, Black or African American 55%, Hispanic or Latino 59%, Mixed Race 100%, Other 22%, Unknown 31%, White or Caucasian 34%
Dopamine D3 Receptor (DRD3): rs6280—risk allele C (Ser9Gly)	41	Asian 56%, Black or African American 93%, Hispanic or Latino 52%, Mixed Race 100%, Other 67%, Unknown 56%, White or Caucasian 54%
Dopamine D4 Receptor (DRD4): rs1800955—risk allele C (48bp repeat VNTR)	42	Asian 44%, Black or African American 57%, Hispanic or Latino 55%, Mixed Race 100%, Other 78%, Unknown 58%, White or Caucasian 70%
µ-Opioid Receptor (OPRM1): rs1799971—risk allele G (A118G)	29	Asian 56%, Black or African American 2%, Hispanic or Latino 28%, Mixed Race 0%, Other 0%, Unknown 39%, White or Caucasian 21%
Serotonin Transporter Receptor (5HTT) Linked Promoter Region (5HTTLPR) in SLC6A4: rs25531—risk allele S’	43 *	Asian 100%, Black or African American 71%, Hispanic or Latino 76%, Mixed Race 100%, Other 56%, Unknown 81%, White or Caucasian 76%

* Data derived from GARS tests of greater than 1000 subjects. ** An rs number is an accession number used by researchers and databases to refer to specific SNPs. It stands for Reference SNP cluster-ID.

## Data Availability

Not applicable.

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
