# Peer review of "Reward Deficiency Syndrome (RDS) Surprisingly Is Evolutionary and Found Everywhere: Is It “Blowin’ in the Wind”?"

_jpm, 2022, doi:10.3390/jpm12020321_

Round 1
Reviewer 1 Report
I am an expert in systematic reviews and meta-analyses. This article can be seen as textbook text or summary of the literature, and is not a systematic review. This text can help readers who do not have time to read all the literature themselves. However, the process of selecting articles is not systematic or transparent but mere a choice of the expert who wrote this.
Author Response
Thank you for the comment. We have changed it to a ‘Perspective’ instead of calling it a ‘Review’.
Reviewer 2 Report
Please find the review report attached.

Reviewer 3 Report
This paper is focused on the Reward Deficiency Syndrome (RDS). This is a review on the different aspects of this syndrome: neurotransmission, prevalence of addiction-related gene polymorphisms, theintegration of the psychological, neurological and biological model, phenomenological aspects and clinical implications. The paper is well written and relevant in the field of mental disorders. However, several changes should be made before publishing it.
In my opinion, the main structure of the review can be improved. The authors are starting about the evolutionary hypothesis of mental illnesses, and afterwards they introduce the genetic basis of RDS, and at the end, the phenomenological aspects are covered.
First of all, after the introduction section, I would recommend to clarify how the authors underwent the review. Which methods did they use? Is it a narrative review? Which were the inclusion and exclusion criteria? Time line of search and search strategy?
I would prefer to divide the neuro-genetic background into several hypothesis: brain structure and function hypothesis? (Here to be clarified the studies reporting PET, and other neuroimaging findings). Genetic hypothesis? Which targets are the authors considering? (e.g. dopaminergic pathways?
Clinical expression or phenomenological aspects should be integrated into the "clinical applications" section.
In summary, the paper is interesting, but the authors should clarify how they did the search and re-structure the sections according to a sequence divided into hypothesis and findings that were tested.
Round 2
Reviewer 1 Report
My main point remains that this is not a research article and certainly not a systematic review. However, in the revised version the authors avoid the words "systematic review".